# Unraveling the Puzzle of Turnover Intention: Exploring the Impact of Home-Work Interface and Working Conditions on Affective Commitment and Job Satisfaction

**DOI:** 10.3390/bs13090699

**Published:** 2023-08-22

**Authors:** Massoud Moslehpour, Afrizal Firman, Jovi Sulistiawan, Pei-Kuan Lin, Hien Thi Thu Nguyen

**Affiliations:** 1Department of Business Administration, Asia University, Taichung 41354, Taiwan; writetodrm@gmail.com (M.M.); afrizalfirman@yahoo.com (A.F.); jovisulistiawan@feb.unair.ac.id (J.S.); congchua_usau1412@yahoo.com (H.T.T.N.); 2Department of Management, California State University, San Bernardino, CA 92407, USA; 3Department of Management, Faculty of Economics and Business, Universitas Airlangga, Surabaya 60115, Indonesia

**Keywords:** turnover intention, affective commitment, job satisfaction, home-work interface, working conditions

## Abstract

This study investigates the antecedents of turnover intention among Vietnamese nurses at a hospital in Quang Ninh Province, North Vietnam. This study evaluates the relationship between home–work interface (HWI) and working conditions (WC) on intrinsic job satisfaction (IJS) and extrinsic job satisfaction (EJS), the relationship between intrinsic job satisfaction (IJS) and extrinsic job satisfaction (EJS) on affective commitment (AC) and turnover intention (TI), and the relationship between affective commitment (AC) and turnover intention (TI). The study employs cross-sectional data and a questionnaire survey to collect the data. The 306 qualified questionnaires were collected, and partial least squares structural equation modeling (PLS-SEM) was employed to analyze the research model and test the hypotheses. The study reveals that working conditions and the home-work interface affect intrinsic job satisfaction. Intrinsic and extrinsic job satisfaction affect affective commitment. Interestingly, affective commitment and intrinsic job satisfaction were not significantly affecting turnover intention. The present study develops and empirically examines a conceptual framework by providing theoretical insight and managerial implications into the turnover intention’s antecedents in Vietnamese nursing care at the hospital.

## 1. Introduction

In industrial and organizational behavior, turnover intention has been seen as a critical issue [1]. Additionally, it is a critical consequence that influences an organization’s performance. Employees’ decision to leave was likely motivated by a desire to work in a position that offered better chances and a desire to experience another ordeal, or possibly due to the challenges they face. The person may wish to quit a difficult work environment to pursue a different career path that aligns with their ambitions. On the other hand, employees will lose their family support if the new work fails to meet their expectations. Thus, it is necessary to consider the fact that not all turnovers are detrimental to the organization. For instance, terminating individuals who perform poorly can assist in placing high achievers in the appropriate positions.

Faheem et al. [2] concluded that a high turnover rate has a detrimental effect on an organization’s efficacy, performance, and even productivity. Thus, turnover affects the effectiveness of communication systems and the performance of work teams. Nurse recruitment and retention will be labeled organizationally compelled if the nursing shortage spreads to the hospital. Due to the high expense of recruiting and training new nurses, a high turnover rate will result in reduced hospital efficiency. Additionally, staff turnover reduces the effectiveness of team-based treatment in inpatient units.

The primary shortfall is that the Vietnamese health sector needs more staff, with 7.7 doctors and 17.6 nurses per 10,000 inhabitants [3]. As a result, the workload of medical workers, particularly nurses, is extremely high to fulfill the increased demand for public health services. The assessment of medical personnel’s health risks and the efficiency of preventative actions will improve their health state, consequently boosting the quality of health services. Numerous Vietnamese hospitals attempt to address this issue by recruiting additional nursing graduates. With such a strong demand for trained nursing personnel, it is vital to research nursing turnover intentions. It will significantly assist hospitals in retaining qualified personnel and providing higher-quality patient care. Healthcare administrators and policymakers must take this into account. Therefore, turnover intention has become a top-priority issue in Vietnam hospitals and the healthcare industry.

This research contributes to a greater understanding of why nurses wish to leave their current hospitals. This understanding will assist hospital administrators and government officials in reducing turnover rates and more effectively managing the turnover system. This study was undertaken in hospitals to ascertain the critical determinants influencing nurses’ decisions to leave or remain in their existing positions. Additionally, it gave an alternate explanation for turnover intentions (TI) by characterizing characteristics such as home–work interface (HWI) and working conditions (WC), extrinsic job satisfaction (EJS), intrinsic job satisfaction (IJS), and affective commitment (AC) as antecedent factors.

The research was conducted at two designated public hospitals in the northern province of Quang Ninh. The hospitals and the two largest in the province were chosen for convenience sampling. The research aids academics, academicians, teachers, and students by broadening their understanding of nurses’ turnover intentions and antecedents. It might serve as a valuable reference for future research. The study’s findings assist health practitioners, managers, and policymakers in determining the most effective approaches, particularly in areas that affect turnover intention.

The research included five chapters, each of which was discussed in depth. Section 1 states the study’s problems, purpose, scope, and contribution. Section 2 consists of a literature review and hypothesis development. Section 3 presents samples, data collection, and measures. Section 4 shows the results of validity, reliability, and hypothesis testing. Section 5 summarizes the discussion of implications, limitations, and future research.

## 2. Literature Review and Hypotheses Development

This section presents the definitions of home–work interface, working conditions, job satisfaction (extrinsic and intrinsic), affective commitment, and turnover intention. The turnover intention antecedents towards employees have been commonly researched in healthcare [1,4]. For instance, turnover intention and job satisfaction studies among healthcare employees in Vietnam, particularly medical workers, and their contributing factors to hospital performance have been investigated by [5]. However, turnover intention and job satisfaction studies on Vietnamese nurses are still relatively new to investigate in the northern province of Quang Ninh Hospital. In addition to investigating the nurses’ turnover intention antecedents, this research hypothesizes the relationship between the home-work interface and working conditions on job satisfaction, job satisfaction on affective commitment and turnover intention, and affective commitment towards turnover intention.

### 2.1. Home–Work Interface (HWI) and Working Conditions (WC)

The home–work interface assesses the extent to which the employer supports employees’ home and family lives [6,7,8]. The home–work interface has an impact on job satisfaction [9]. Many studies have proven that the home–work interface can significantly influence nurses’ job satisfaction [10,11,12,13]. Employers may wish to consider methods to assist nurses in managing the home–work interface to enhance job satisfaction and performance. On the other hand, when employees can effectively balance their work and family responsibilities and experience positive spillover from work into their personal lives, this can lead to greater job satisfaction [9]. It reflects that HWI will affect nurses’ ability to implement their tasks effectively, which impacts their job and career satisfaction. The authors wish to examine the effect of the HWI on extrinsic and intrinsic job satisfaction. Thus, the present study proposes the following hypotheses:

**H1.** 
*HWI has a positive effect on EJS.*


**H2.** 
*HWI has a positive effect on IJS.*


On the other hand, working conditions represent how satisfied employees are with their work environment and the level of security required to do their responsibilities properly [6,7,8]. Therefore, a healthy WC can reduce nurses’ turnover, hence job satisfaction, and lower job stress [14]. Arıkan Dönmez et al. [15] found that positive WC resulted in more motivated nurses and an increase in the efficacy of the hospital. Magnavita et al. [16] argued that improving healthy WC in healthcare workers affects job satisfaction. Supportively, the studies of WC, such as salary, working environment, psychological empowerment, and relationships with supervisors and colleagues, have evidenced a significant effect on the job satisfaction of nurses in Vietnam [17,18,19]. Thus, the present study proposes the following hypotheses:

**H3.** 
*WC has a positive effect on EJS.*


**H4.** 
*WC has a positive effect on IJS.*


### 2.2. Intrinsic Job Satisfaction (IJS) and Extrinsic Job Satisfaction (EJS)

Hirschfeld [20] showed that job satisfaction is divided into two dimensions: intrinsic job satisfaction (IJS) and extrinsic job satisfaction (EJS). Buitendach and Witte [21] defined that IJS includes job content and tasks such as skill usage, autonomy, self-fulfillment, and self-growth. In the healthcare sector, job satisfaction can improve nurses’ commitment levels, thus diminishing their TI [22,23]. IJS also impacts nurses’ decisions to stay or leave the hospital. If a nurse thinks that the hospital treats him (or her) so well that (s)he can get what (s)he deserves, (s)he will choose to stay at the hospital. The other literature shows that job satisfaction significantly affects commitment [24]. Nurses with great job satisfaction will have positive outcomes on AC and TI [25,26]. Thus, the present study proposes the following hypotheses:

**H5.** 
*IJS has a positive effect on AC.*


**H6.** 
*IJS has a positive effect on TI.*


Extrinsic job satisfaction refers to a hospital’s motivation for its employees, such as salary, opportunities for advancement or growth, and the working environment [21]. In the healthcare industry, there are several reasons why nurses are unsatisfied with their jobs, including a lack of engagement in decision-making, poor communication with management, low earnings, inadequate benefits, and a lack of scheduling flexibility [22,23,27]. Additionally, the study discovered that the primary source of unhappiness and intention to leave the hospital was strongly tied to the institution’s incentives and compensation, referred to as extrinsic rewards.

Mbah and Ikemefuna [28], Sharma and Dhar [26], and Falatah and Conway [25] highlighted that greater job satisfaction results in a lower likelihood of intention to leave. Job satisfaction means that a nurse who is satisfied with their employees will have a positive attitude and AC toward them. In conclusion, satisfied nurses will remain at their existing hospitals. Therefore, the hypothesis suggests a significant correlation between EJS’s attitude towards AC and TI. Thus, the present study proposes the following hypotheses:

**H7.** 
*EJS has a positive effect on AC.*


**H8.** 
*EJS has a positive effect on TI.*


### 2.3. Affective Commitment (AC)

Allen and Meyer [29] defined affective commitment as an emotional commitment to the organization in which the strongly committed individual identifies with, participates in, and benefits from participation. AC refers to nurses’ identification with and emotional attachment to their hospitals [30,31,32]. Nurses’ positives include increased enthusiasm, stability, and belongingness [33]. Nurses with a strong AC tend to work harder and have a strong emotional ambition to stay with hospitals to enhance hospital performance [33]. Akinyemi et al. [34] found that the shortage of AC leads to an increase in turnover rate and TI. However, AC is one of the best predictors of TI [35,36]. Similarly, Ito et al. [37] emphasized that AC has a key role in preventing TI on hospital nurses’ retention in Japan. Thus, the present study proposes the following hypothesis:

**H9.** 
*AC has a positive effect on TI.*


## 3. Method

### 3.1. Sample and Data Collection

We employed the recommendation from Hair et al. [38] in determining the minimum sample for this study. Our study has two independent variables, we set the significance level at 5% and the minimum R^2^ at 0.1; the minimum sample size is 90 with 80% statistical power. The current study enrolled 400 nurses from two public hospitals in Quang Ninh province in northern Vietnam who were invited to participate. The huge sample size means that quantitative research will be used to generate statistically credible and accurate results. The response rate was 76.5 percent. We contacted the hospital’s human resources department to aid with survey administration. At the first meeting, we presented and described the goal of the current research and its significance in contributing to the efficacy and benefits of their hospitals. We enlisted their cooperation in delivering questionnaires to nurses after receiving their assent. The respondents were instructed on how to complete the survey.

Before completing the questionnaire, all respondents were asked to sign an informed consent form to confirm that they were voluntarily participating in the study and were aware of its objectives, significance, advantages, and potential dangers associated with their involvement. They received no promotions or benefits in exchange for completing the questionnaire. We collected the documents each week and maintained contact with hospital human resources personnel. Thirty-six copies were returned. The survey tool’s items were originally written in English and then translated into Vietnamese for this investigation. The individuals’ demographic data were used as control variables. The items contain the following information in Table 1: gender, age, education, marital status, tenure, monthly income, and overtime days off.

For the gender frequencies, males were 65 people (21.2%), and females were 241 people (78.8%). The gap between males and females was quite big. The majority of nurses were female. For the age frequencies, there were five categories represented in terms of age: less than 21 years old (30 people = 9.8%), 21–30 years old (95 people = 31%), 31–40 years old (73 people = 23.9%), 41–50 years old (52 people = 17.0%), and more than 51 years old (56 people = 18.3%). Most of the respondents were aged 21 to 30 years old. Education frequencies were in four categories: high school, bachelor’s, master’s, and Ph.D. Most respondents had bachelor’s degrees (265 people = 86.6%). Marital status frequencies were categorized into two items: married (235 people = 76.8%) and single, including divorced and widowed (71 people = 23.2%).

Tenure at the current hospital frequency was categorized into five characteristics. The most frequent response was 2–5 years (132 people = 43.1%); respondents who worked for more than ten years occupied 7.8%. Monthly income frequencies were three categories: most respondents had a monthly income of USD 100–USD 200 (243 people = 79.4%). In summary, the Vietnamese nurses’ income in the present study was very low. Overtime in a day off frequency was 250 nurses (81.7%) who worked overtime.

### 3.2. Measures

The participants must respond to those items on a five-point Likert scale, where five indicates “strongly agree”, and one indicates “strongly disagree”. High scores represent a high level of agreement. We developed a questionnaire based on our review of the literature. HWI and WC were measured using the three-item instrument [8]. IJS and EJS were measured using the five-item instrument [39,40]. AC was measured using six items [33]. TI was measured using the four item instrument [41]. Furthermore, the measurement items are described in Table 2.

## 4. Results

This section summarizes the validity, reliability, and hypothesis testing of the study, which were analyzed using structural equation modeling (SEM) [42,43].

### 4.1. Validity and Reliability

Cronbach’s alpha coefficient (α), composite reliability (CR), and average variance extracted (AVE) are used to indicate the instrument’s dependability. Cronbach’s alpha and CR values are higher than the usual threshold of 0.70. In comparison, AVE values are higher than the needed minimum of 0.50, and the composite reliability score is considered reliable if greater than 0.70. Each item had an outer loading greater than 0.50. As a result, all necessary indices are satisfactorily greater than the minimum required values (Table 3 and Figure 1). Furthermore, all values range between 0.062 and 0.696, not exceeding the threshold for the heterotrait–monotrait correlation ratio (HTMT) criterion as attached in Table 4 and each variable’s AVE score is greater than the correlation coefficient for all other variables, as seen in Table 4 of the Fornell–Larcker criterion [44]. The evaluations indicate that all the measurement models demonstrate convergent and discriminant validity.

### 4.2. Assessment of Structural Model

The first step in evaluating the structural model is to address any collinearity problems that have been raised. Given that the VIF values of all of the predictor constructs were significantly below the 3.3 threshold (ranging from 1.005 to 1.179), as determined by the investigation, this suggests that collinearity was not a problem with this study [38,45]. Next, we assessed the significance of the predicted associations by employing the bootstrapping method on 5000 subsamples.

The path coefficients were assessed to test our hypotheses. As a result, there is no statistically significant association between HWI and EJS (β = 0.049, N.S.). However, there is a substantial positive correlation between HWI and IJS (β = 0.174, *p* < 0.008). Positive correlations exist between W.C. and EJS (β = 0.574, *p* < 0.000) and IJS (β = 0.120, *p* < 0.023). IJS has a significant positive effect on A.C. (β = 0.373, *p* < 0.000) but does not affect T.I. (β = −0.017, N.S.). Two hypotheses were statistically significant: one about the association between EJS and A.C. (β = 0.088, *p* < 0.047), and another about the relationship between EJS and T.I. (β = 0.523, *p* < 0.000). Finally, A.C. did not affect T.I. (β = 0.049, N.S.). Table 5 and Figure 2 summarize the connection and path coefficients’ outcomes.

The predictive accuracy of the model was then evaluated using Cohen’s standards for the coefficient of determination (R2), which are 0.02 (weak), 0.13 (moderate), and 0.26 (substantial) [46]. As can be seen in Figure 2, the model displayed a significant degree of predictive accuracy, with R2 values of 0.348 for EJS, 0.033 for IJS, 0.0152 for AC, and 0.280 for TI, respectively. Additionally, the model displayed good predictive relevance, with Q2 values that ranged from 0.185 (turnover intention) to 0.019 (intrinsic job satisfaction), all of which were significantly higher than zero. In addition, we also assessed the standardized root mean square residual (SRMR) in order to determine the absolute measure of model fit. The SRMR score in our study is 0.066, which complies with the minimum threshold suggested by Henseler et al. that SRMR scores less than 0.08 are considered a good fit [44].

## 5. Discussions

The current research studies the antecedents of turnover intention among Vietnamese nurses in Quang Ninh Province, North Vietnam, hospitals. The study investigated three objectives: evaluating the relationship between HWI and WC factors on IJS and EJS, the relationship between IJS and EJS factors on AC and TI, and the relationship between AC and TI. The results statistically indicate that there is no significant support between HWI and EJS, and there is significant support between HWI and IJS. Meanwhile, WC has full support for both EJS and IJS. Furthermore, IJS has significant support for AC but not significant support for TI. On the other hand, EJS has significant support for both AC and TI. However, AC’s relationship with TI needs to be supported. These study hypotheses were statistically analyzed by a quantitative tool, PLS-SEM, and comprehensively reported in this discussion part.

### 5.1. Theoretical and Managerial Implications

The research contributes to our understanding and insights into the HWI and WC as independent variables, their relationship to EJS and IJS as mediating variables, and the relationship of EJS and IJS to AC and TI as dependent variables. Based on our evidence, the present study develops and empirically examines a conceptual framework that provides theoretical insight into turnover intention’s antecedents in Vietnamese nursing care at the hospital. We build a theoretical framework and hypotheses based on the previous studies to answer the research objectives.

Three objectives of the study are tested and found to have theoretical implications. Thus, the findings of the study offer several theoretical implications. First, our study revealed that both intrinsic and extrinsic job satisfaction are significantly affected by WC. Nurses are a professional cohort that assumes significant responsibilities and faces demanding work requirements. Enhancements in working circumstances have an impact on their job satisfaction. Furthermore, our study suggests that a significant determinant of job satisfaction is the quality of working conditions. It is consistent with prior studies that show health-care workers’ job satisfaction is significantly affected by working conditions, especially in developing countries [47]. Moreover, while enthusiasm serves as a motivating factor that elicits fulfillment among nurses, a significant proportion of nurses in developing countries choose to prioritize their working conditions due to the substantial risks they encounter.

Second, the home–work interface does not affect nurses’ extrinsic job satisfaction. This result is contrary to most prior studies. The lack of significance in the obtained result can be attributed to cultural factors [48]. The impact of cultural values, specifically individualism, and collectivism, on the extent to which the HWI affects extrinsic job satisfaction is significant. Employees in countries with a strong individualistic orientation prefer to prioritize their personal needs, which may explain their tendency to behave more negatively when their employment responsibilities encroach upon these demands. Employees in countries with lower levels of individualism, such as Vietnam, are more likely to perceive such interferences as inherent to their role as nurses. Furthermore, individuals residing in a collectivistic community may exhibit a higher propensity to maintain their commitment to their occupation [49].

Third, the home–work interface positively affects nurses’ intrinsic job satisfaction. Our result confirms prior studies that indicate employers’ efforts to support employees’ work-life balance are crucial in enhancing nurses’ intrinsic job satisfaction [9,50]. In contrast to our previous findings, which suggested that HWI does not have a substantial impact on predicting extrinsic motivation, the association between HWI and intrinsic job satisfaction is consistent across diverse contexts with varying cultural values and assumptions, including the context of Vietnam. Moreover, our findings suggest that when employers provide help to nurses in managing their work and family responsibilities, it affords them the opportunity to utilize a diverse set of professional skills and abilities [51].

Fourth, our results revealed that IJS and AC do not significantly affect TI. These results are interesting since most prior studies showed that IJS and AC became the critical factors in predicting employees’ turnover intentions [25,26]. The reason for this outcome can be attributed to the nurse’s professional occupation. In contrast to occupations such as salespersons, which often allow for easy occupational transitions, nurses tend to exhibit a greater degree of commitment to their profession than to specific employers or organizations. Previous research has indicated that organizational commitment emerges as a very influential factor. However, when considering professional occupations, the impact of organizational commitment is comparatively weaker in comparison to professional dedication [52].

This study has some managerial implications that can benefit both the public and private sectors of the Vietnamese healthcare system, particularly hospitals. First, this research advances our understanding of why nurses wish to quit their existing positions in hospitals. Second, the research assists hospital administrators and government officials in lowering turnover rates and operating costs and improving the turnover system’s management. Third, the findings assist healthcare practitioners, managers, and policymakers in determining the most effective solutions, particularly in the areas that have the greatest effect on turnover intention. Fourth, this empirical study found that nurses’ affective commitment does not significantly impact the hospital turnover intention rate.

Vietnamese hospitals should search for effective strategies for lowering employee dissatisfaction. According to the findings, managers in the healthcare industry can reduce employee turnover by using some of the suggested approaches. Hospital administrators can provide nurses with more flexibility in their schedules. Providing nurses with proper sleep and rest will improve their working conditions. On the other hand, managers are responsible for maintaining an ordered and safe work environment and addressing appropriate nurse concerns. Additionally, they may explore increasing nursing staff motivation through remuneration, bonuses, and other incentives for nurses who perform well on the job.

### 5.2. Limitations and Future Research

Despite its significant contributions, this study has several limitations. First, the surveys should be provided to nursing professionals in both public and private institutions for future research. It should be spread throughout all hospitals in Vietnam, not only those in the north. Second, this study used a quantitative method and self-administered structural questions to investigate the desire of Vietnamese nursing workers to leave. Qualitative methodologies utilizing open-ended questions, such as telephone surveys and interviews, can be utilized to obtain accurate data in the future. Third, with regard to our contrasting findings, future studies may employ intervening variables such as professional or career commitment that may serve as the mediating mechanisms of AC and TI. Last, future studies may consider cultural values by comparing two countries that have different cultural values, such as high-individualized compared to high-collectivist countries.

## 6. Conclusions

This study investigated the antecedents of turnover intention (TI) towards Vietnamese nurses at the hospital. It evaluated three objectives: the relationship between home–work interface (HWI) and working conditions (WC) on intrinsic job satisfaction (IJS) and extrinsic job satisfaction (EJS), the relationship between intrinsic job satisfaction (IJS) and extrinsic job satisfaction (EJS) on affective commitment (AC) and turnover intention (TI), and the relationship between affective commitment (AC) and turnover intention (TI). However, the study employed SEM to analyze the research model and test the hypotheses. The research findings contribute to our understanding of the effects of home–work interface (HWI) and working conditions (WC) on nurses’ job satisfaction, which consists of intrinsic and extrinsic job satisfactions. In addition, our results revealed that intrinsic job satisfaction (IJS) and extrinsic job satisfaction (EJS) were significant predictors of affective commitment (AC). Contrary to our hypotheses, the results showed that affective commitment (AC) and intrinsic job satisfaction (IJS) were not significant in predicting turnover intention (TI). Our findings validate prior studies showing that organizational commitment was not a strong predictor for professional occupations, such as nursing. The findings of this study assist hospitals in determining the most effective approaches, particularly in areas identified as influencing turnover intention.

## Figures and Tables

**Figure 1 behavsci-13-00699-f001:**
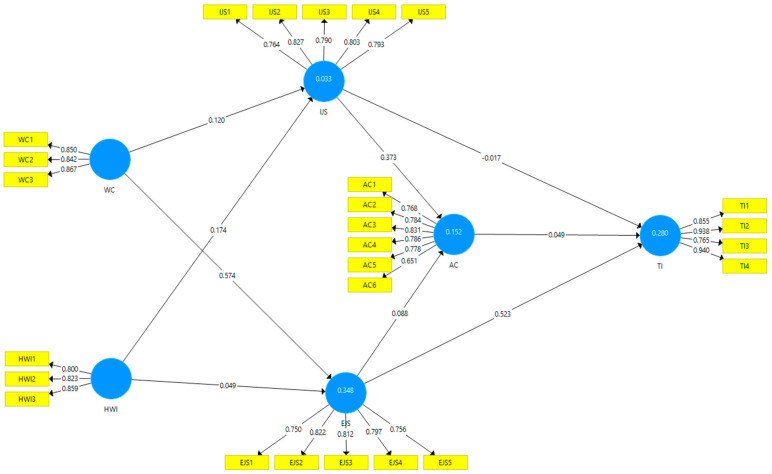
PLS Structural Path.

**Figure 2 behavsci-13-00699-f002:**
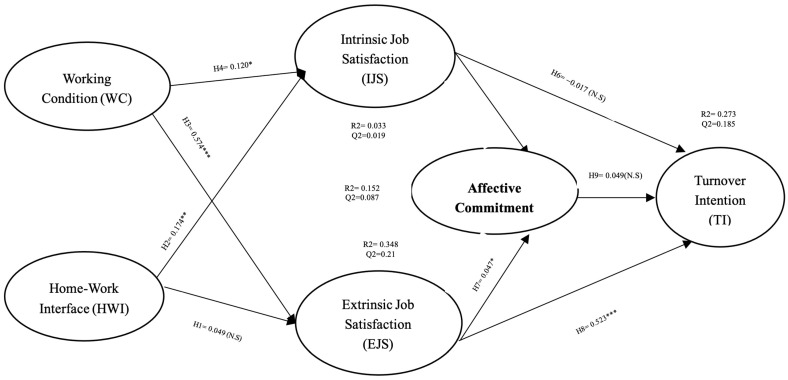
The results of structural equation modeling. **Note(s):** * *p* < 0.05; ** *p* <0.001; *** *p* < 0.001; N.S = Not Significant.

**Table 1 behavsci-13-00699-t001:** Demographic Characteristics.

Profile	Classifications	Samples (N = 306)	Percentage (%)
Gender	Male	65	21.2
	Female	241	78.8
Age	<21	30	9.8
	21–30	95	31.0
	31–40	73	23.9
	41–50	52	17.0
	>50	56	18.3
Education	High school	41	13.4
	Bachelor	265	86.6
	Master	0	0
	Ph.D.	0	0
Marital status	Single (including divorced and widowed)	71	23.2
	Married	235	76.8
Tenure	<1 year	43	14.1
	1–2 years	75	24.5
	2–5 years	132	43.1
	5–10 years	32	10.5
	>10 years	24	7.8
Monthly income	<$100	26	8.5
	$100–$200	243	79.4
	>$200	37	12.1
Overtime day off	Frequently	250	81.7
	Sometime	38	12.4
	Rarely	11	3.6
	Never	7	2.3

**Table 2 behavsci-13-00699-t002:** Measurement items.

No.	Variables	Measurements	References
1	HWI	In most ways, my life is close to ideal	Van Laar, et al. [8]
		I work in a safe environment	
		Generally, things work out well for me	
2	WC	Recently, I have been feeling unhappy and depressed because my working conditions are not performing well that effecting to my skills and career	
		I am satisfied with my life when my working conditions significantly affect on my career development	
		I am encouraged to develop new skills	
3	IJS	The chance to work alone on the job	Schriesheim, et al. [39]
		The chance to do things for other people	
		The chance to do something that makes use of my abilities	
		The chance to try my own methods of doing the job	
		The feeling of accomplishment I get from the job	
4	EJS	The working conditions satisfy my job performance	Martins [40]
		The praise I get for doing a good job	
		The way hospital’ policies are put into practice	
		My pay and the amount of work I do	
		The chances for advancement in this job	
5	AC	I would be very happy to spend the rest of my career in this hospital	Meyer, et al. [33]
		I do not feel like ‘part of my family’ at this hospital	
		I do not feel ‘emotionally attached’ to this hospital	
		This hospital has a great deal of personal meaning for me.	
		I do not feel a strong sense of belonging to this hospital	
		I think that I could easily become as attached to another organization as I am	
6	TI	As soon as I can find a better job, I will quit this hospital	Olusegun [41]
		I often think about quitting my job	
		I will likely actively look for a new job in the next year	
		It is very unlikely that I would ever consider leaving this hospital	

**Table 3 behavsci-13-00699-t003:** The validity and reliability results.

Variables	Items	Outer Loadings	*α*	CR	AVE
Affective Commitment (AC)	AC1	0.768	0.860	0.896	0.590
	AC2	0.784			
	AC3	0.831			
	AC4	0.786			
	AC5	0.778			
	AC6	0.651			
Extrinsic Job Satisfaction (EJS)	EJS1	0.750	0.848	0.891	0.621
	EJS2	0.822			
	EJS3	0.812			
	EJS4	0.797			
	EJS5	0.756			
Home-Work Interface (HWI)	HWI1	0.800	0.771	0.867	0.685
	HWI2	0.823			
	HWI3	0.859			
Intrinsic Job Satisfaction (IJS)	IJS1	0.764	0.855	0.896	0.633
	IJS2	0.827			
	IJS3	0.790			
	IJS4	0.803			
	IJS5	0.793			
Turnover Intention (TI)	TI1	0.855	0.901	0.930	0.770
	TI2	0.938			
	TI3	0.765			
	TI4	0.940			
Working Condition (WC)	WC1	0.850	0.814	0.889	0.728
	WC2	0.842			
	WC3	0.867			

**Table 4 behavsci-13-00699-t004:** The Fornell-Larcker Criterion and HTMT *.

	AC	EJS	HWI	IJS	TI	WC
1. AC	**0.768**	*0.145*	*0.099*	*0.437*	*0.106*	*0.176*
2. EJS	0.114	**0.788**	*0.255*	*0.093*	*0.558*	*0.696*
3. HWI	−0.049	0.209	**0.828**	*0.170*	*0.165*	*0.347*
4. IJS	0.380	0.070	−0.140	**0.796**	*0.062*	*0.093*
5. TI	0.102	0.527	0.144	0.038	**0.877**	*0.380*
6. WC	0.148	0.588	0.278	0.072	0.351	**0.853**

* Note(s): AC—affective commitment; EJS—extrinsic job satisfaction; HWI—home-work interface; IJS—intrinsic job satisfaction; TI—turnover intention; WC—working condition. The values in **bold** on the diagonal represent the square root of AVE. The values in *italic* above the diagonal values represent the heterotrait–monotrait (HTMT) scores.

**Table 5 behavsci-13-00699-t005:** The results of SEM.

Hypotheses	Path	Original Sample (O)	Standard Deviation (STDEV)	T Statistics (|O/STDEV|)	*p* Values	Remarks *
H1	HWI → EJS	0.049	0.053	0.937	0.175	n.s
H2	HWI → IJS	0.174	0.071	2.437	0.008	supported
H3	WC → EJS	0.574	0.058	9.924	0.000	supported
H4	WC → IJS	0.120	0.060	1.993	0.023	supported
H5	IJS → AC	0.373	0.058	6.453	0.000	supported
H6	IJS → TI	−0.017	0.054	0.315	0.376	n.s
H7	EJS → AC	0.088	0.052	1.681	0.047	supported
H8	EJS → TI	0.523	0.050	10.397	0.000	supported
H9	AC → TI	0.049	0.063	0.768	0.222	n.s

* **Note(s):** AC—affective commitment; EJS—extrinsic job satisfaction; HWI—home-work interface; IJS—intrinsic job satisfaction; TI—turnover intention; WC—working condition; n.s.—Not Sig.

## Data Availability

Not applicable.

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
