# Peer review of "Unraveling the Puzzle of Turnover Intention: Exploring the Impact of Home-Work Interface and Working Conditions on Affective Commitment and Job Satisfaction"

_behavsci, 2023, doi:10.3390/bs13090699_

Round 1
Reviewer 1 Report
The manuscript has potential, but there are numerous aspects that require improvement.
1. Based on an analysis of the research model and its constituent constructs, it can be inferred that the title of the manuscript does not align with the paper's content or the underlying research model. What happened to affective commitment (AC) and turnover intention (TI)?
2. In the abstract, the authors stated that they used structural equation modeling (SEM), but actually the PLS-SEM approach was used.
3. The literature review does not sufficiently establish a comprehensive foundation for the formulated hypotheses. The authors state "The home-work interface assesses the extent to which the employer supports employees' home and family lives (Ali Mohammad, 2013; C. Fradelos et al., 2022; Van Laar et al., 2007). When the negative HWI becomes significant, the only solution for nurses is to leave the hospital and reduce work demands that affect family life (Rode et al., 2007). It reflects that HWI will motivate nurses to implement their tasks effectively, which impacts their job career satisfaction. The authors wish to examine the effect of the HWI on extrinsic job satisfaction and intrinsic job satisfaction." and then state the following hypotheses:
H1: HWI has a positive effect on EJS
H2: HWI has a positive effect on IJS
This definition of hypotheses is not supported by the prior review of the relevant literature. How was the conclusion that "HWI will motivate nurses to implement their tasks effectively" reached? How does effectiveness relate to motivation? he authors should conduct a more detailed review of the literature as a basis for the hypotheses. Also, authors should strictly take care of the use of each term in a sentence, because there are close terms that do not have the same or at least a similar meaning. This should be applied to the literature review that precedes all hypotheses.
4. By what criteria was the sample size determined? What was the rationale behind the decision to include 400 nurses in the study? What were the criteria used to make the decision?
5. The following text "Error! Reference source not found". It is unclear how this error occurred and what it means. It is certain that this should be corrected throughout the work.
6. The following statements were used for the construct "Working conditions":
"Recently, I have been feeling unhappy and depressed"
"I am satisfied with my life"
"I am encouraged to develop new skills"
In what way can the above statements form the construct "Working conditions"? It is unclear how "I am satisfied with my life" is related to working conditions. The item "Recently, I have been feeling unhappy and depressed" seems very far from being able to measure working conditions.
7. In the part related to the validity and reliability analysis, it is necessary to provide data for the value of the VIF coefficient. The proposed research model lacks a statistical evaluation of the model itself. What is the value of the Stoner-Geisser Q2, R2, GOF, and SRMR coefficients for the entire model?
8. It remains unclear why the authors used structural equation modeling. SEM or PLS-SEM are usually used when there is an intention to investigate the indirect effect between variables (mediation, moderation). It was expected that the mediating effect would be investigated, because in that case the title "Unraveling the Puzzle of Turnover Intention..." would make full sense.
The authors should decide whether their model will have only direct effects or whether they will also include indirect effects in the analysis. In both cases, the following claims, which cannot find support in the given statistical analysis, should be reconsidered.
In Discussion section: "The research contributes to our understanding and insights of the HWI and WC as independent variables on the relationship to EJS and IJS as mediating variables and the relationship of EJS and IJS on AC and TI as dependent variables." and "First , WC is the most significant influence on the relationship between HWI and WC with EJS and IJS."
In Conclusion section:
"The research findings contribute to our understanding and insights of the HWI and WC as independent variables on the relationship to EJS and IJS as mediating variables and the relationship of EJS and IJS on AC and TI as dependent variables."
There is no statistical analysis in the paper that confirms the previous statements.
Author Response
Dear reviewer,
I am writing to provide an update on the revision process of our paper [Paper id:behavsci-2515669], which you graciously reviewed. We sincerely appreciate the time and effort you invested in providing valuable feedback, as it has been instrumental in improving the quality and of our work.
First, we want to express our gratitude for the insightful suggestions you made. Your comments have guided us in refining our research and enhancing the clarity of our findings. We carefully considered each of your recommendations and diligently worked towards addressing them in the revised version. Please kindly see the attachment for our response.
We believe that these revisions have addressed the concerns raised during the review process. However, if you feel that there are still areas that require further attention, we would be more than willing to make additional adjustments to ensure the paper meets the highest standards.

Reviewer 2 Report
The topic is interesting and offer indepth insights in the issue under study.
There are some intext citation errors, such as line 87 ((Ali Mohammad, 2013; C. Fradelos et al., 2022; Van Laar et 87 al., 2007).), where two names of first author are taken and initial of another author is taken. Please correct these and all other errors. this mistake is also repeated in line 98.
Another example of intext citation error is on line 170 (n Error! Reference source not found. ) please correct this and other similar errors appearing multiple times.
The literature review section does not offer insights into the research gap to justify this research. So add a paragraph on existing research gap within this research context.
The discussion section is too brief and without any critical and comparative insights. These results should be discussed in relation to the previous studies to offer a comparative analysis of this study with previous ones.
in limitations section add at least one more point related to culture/context
Author Response

(The authors gave the same response as above.)

Round 2
Reviewer 1 Report
The most important parts of the work have been adequately corrected. There are two points that require special attention:
1. The authors wrote "First, our study revealed that WC is crucial in predicting nurses' job satisfactions. It suggests that the health industry is typically the most impacted by working conditions."
Have authors included and studied other factors that influence job satisfaction? The use of the word "crucial" in this context is questionable.
2. The authors wrote "Third, with regard to our contrasting findings, future studies may employ moderating variables such as professional commitment that may strengthen or weaken the relationship between AC and TI."
It is recommended to use "intervening" rather than "moderating" because it is possible to receive "only" a mediating effect instead of a moderating effect.
Author Response
Dear reviewer,
We are writing to provide an update on the revision process of our paper which you graciously reviewed. We sincerely appreciate the time and effort you invested in providing valuable feedback, as it has been instrumental in improving the quality and impact of our work.
First and foremost, we want to express our gratitude for the insightful suggestions you made. Your comments have guided us in refining our manuscript. We carefully considered each of your recommendations and diligently worked towards addressing them in the revised version of the paper.
We believe that these revisions have significantly strengthened the paper and addressed the concerns raised during the review process. However, if you feel that there are still areas that require further attention, we would be more than willing to make additional adjustments to ensure the paper meets the highest standards. Our detail response is attached.
Thank you once again for your invaluable contribution to the improvement of our research. We eagerly await your feedback on the revised manuscript and look forward to hearing from you soon.
Please feel free to reach out to us if you have any further suggestions or queries. Your expertise and insights are highly respected and appreciated.
Regards.
